# Therapeutic Advances and Challenges for the Management of HPV-Associated Oropharyngeal Cancer

**DOI:** 10.3390/ijms25074009

**Published:** 2024-04-03

**Authors:** Isis de Araújo Ferreira Muniz, Megan Araujo, Jenna Bouassaly, Fatemeh Farshadi, Mai Atique, Khashayar Esfahani, Paulo Rogerio Ferreti Bonan, Michael Hier, Marco Mascarella, Alex Mlynarek, Moulay Alaoui-Jamali, Sabrina Daniela da Silva

**Affiliations:** 1Department of Otolaryngology and Head and Neck Surgery, McGill University, Montreal, QC HC3 1E2, Canada; isisamuniz@gmail.com (I.d.A.F.M.); fatemeh.farshadi@mail.mcgill.ca (F.F.); mai.atique@mail.mcgill.ca (M.A.); pbonan@yahoo.com (P.R.F.B.); michael.hier@mcgill.ca (M.H.); marco.mascarella@mcgill.ca (M.M.); alex.mlyanarek@mcgill.ca (A.M.); moulay.alaoui-jamali@mcgill.ca (M.A.-J.); 2Graduate Program in Dentistry, Federal University of Paraíba, João Pessoa 58051-900, PB, Brazil; 3Division of Experimental Medicine and Oncology, Department of Medicine and Health Sciences, McGill University, Montreal, QC HC3 1E2, Canada; megan.araujo@mail.mcgill.ca (M.A.); jenna.bouassaly@mail.mcgill.ca (J.B.); 4Department of Oncology, McGill University, Montreal, QC HC3 1E2, Canada; khashayar.sfahani@mcgill.ca

**Keywords:** head and neck cancer, oropharyngeal cancer, clinical trial, chemotherapy, immunotherapy, drug discovery

## Abstract

The use of conventional chemotherapy in conjunction with targeted and immunotherapy drugs has emerged as an option to limit the severity of side effects in patients diagnosed with head and neck cancer (HNC), particularly oropharyngeal cancer (OPC). OPC prevalence has increased exponentially in the past 30 years due to the prevalence of human papillomavirus (HPV) infection. This study reports a comprehensive review of clinical trials registered in public databases and reported in the literature (PubMed/Medline, Scopus, and ISI web of science databases). Of the 55 clinical trials identified, the majority (83.3%) were conducted after 2015, of which 77.7% were performed in the United States alone. Eight drugs have been approved by the FDA for HNC, including both generic and commercial forms: bleomycin sulfate, cetuximab (Erbitux), docetaxel (Taxotere), hydroxyurea (Hydrea), pembrolizumab (Keytruda), loqtorzi (Toripalimab-tpzi), methotrexate sodium (Trexall), and nivolumab (Opdivo). The most common drugs to treat HPV-associated OPC under these clinical trials and implemented as well for HPV-negative HNC include cisplatin, nivolumab, cetuximab, paclitaxel, pembrolizumab, 5-fluorouracil, and docetaxel. Few studies have highlighted the necessity for new drugs specifically tailored to patients with HPV-associated OPC, where molecular mechanisms and clinical prognosis are distinct from HPV-negative tumors. In this context, we identified most mutated genes found in HPV-associated OPC that can represent potential targets for drug development. These include *TP53*, *PIK3CA*, *PTEN*, *NOTCH1*, *RB1*, *FAT1*, *FBXW7*, *HRAS*, *KRAS,* and *CDKN2A*.

## 1. Introduction

Head and neck mucosal cancer (HNC) involves a heterogeneous group of malignant tumors that can affect different sites of the oral cavity, pharynx, and larynx [1,2,3]. The prevalence of these cancers has increased during the last 30 years [1,2,3]. The main risk factors associated with HNC are alcohol and tobacco consumption, followed by human papillomavirus (HPV) infection [1,2,4], which significantly impacts the patient’s prognosis [5,6,7]. Treatment plans are based on the clinical and pathological stage of the cancer and consist of surgery, radiation therapy, chemotherapy, immunotherapy, or a combination of these treatments’ modalities [8,9] (Figure 1). Surgery is the primary treatment for most resectable oral cancers as well as many larynx cancers. Most tumors in the head and neck region are diagnosed at advanced stages, where chemotherapy combined with radiotherapy are standard treatment. However, this treatment approach is associated with toxicity and severe side effects [10,11,12]. At present, patients with HNC have one of the lowest survival rates among cancer patients despite recent advances in therapeutic discovery [2,13].

New therapeutic approaches have been developed in an attempt to improve efficacy while mitigating the undesirable side effects of chemotherapy and chemoradiation and improving the quality of life of patients [2,10]. Several targeted chemo- and immune-therapeutics have been integrated into de-intensified strategies to improve or maintain response rates while minimizing treatment-related morbidities. Due to its unique etiology and superior prognosis, patients with HPV-positive oropharyngeal cancer (OPC), the most common HPV-associated HNC, may benefit from de-escalated strategies. Currently, eight semi-synthetic or synthetic agents have been approved for use against all HNC subtypes, including bleomycin sulfate, cetuximab (Erbitux), docetaxel (Taxotere), hydroxyurea (Hydrea), pembrolizumab (Keytruda), loqtorzi (Toripalimab-tpzi), methotrexate sodium (Trexall), and nivolumab (Opdivo) [14]. Furthermore, based on a systematic data collection from the literature, our team has recently identified commonly mutated genes in HPV-positive OPC, which may provide avenues for novel therapeutic target selection and drug development [15]. This comprehensive review explored the status of approved therapies (chemotherapy, immunotherapy, and target therapy), therapies that are currently under investigation, and potential investigational drugs and treatment strategies that can be further studied in the context of head and neck cancer research.

## 2. Materials and Methods

This study did not require ethical approval or informed consent, as the analyses were carried out based on data from previously published clinical trials and the published literature.

### 2.1. Literature Search

This review was carried out through searches in the PubMed/Medline (1946 to 2023), Scopus, and International Statistical Institute (ISI) web of science databases. Briefly, the search included keywords and mesh terms such as “head and neck cancer”, “oropharyngeal cancer”, “chemotherapy”, “drugs”, “treatment”, “chemoradiotherapy”, and “pharmacotherapy”.

### 2.2. Clinical Trials Selection: Inclusion and Exclusion Criteria

The data from the clinical trials were extracted from the World Health Organization (WHO), International Clinical Trials Registry Platform (ICTRP) (http://apps.who.int/trialsearch/ accessed on 1 February 2024), Current Controlled trials (www.controlledtrials.com/ accessed on 1 February 2024), and Clinical Trials (www.clinicaltrials.gov/ accessed on 1 February 2024).

Filters were applied to select the interventional clinical trials considered in “recruiting”, “not recruiting”, “active, not recruiting”, and “applying by invitation”. The search was performed until 21 November 2023. Clinical trials whose primary objective was treatment, which evaluated the use of drugs combined or not with radiotherapy and immunotherapy, were included. Clinical trials focused only on prevention, supportive care, basic science, behavior, diagnosis, nutritional/supplemental treatment, and radiotherapy alone were excluded. Data extraction included the drugs used for the chemotherapy (CT), the NCT number (number of the register), clinical trial status, HPV status, clinical intervention, clinical phase, population, the date that the study started and was completed, and the country.

### 2.3. Genes Involved in HPV-Associated OPC

Based on our recent publication compiling data from 38 studies retrieved from four databases (Medline, PubMed, Web of Science, and Scopus), we identified the most cited genes in relation to HPV-associated OPC. These studies spanned 8311 patients across 12 countries. The mutated genes identified most often in these 38 studies included *TP53* (*n* = 22), *PIK3CA* (*n* = 20), *PTEN* (*n* =16), *NOTCH1* (*n* = 14), *RB1* (*n* = 13), *FAT1* (*n* = 13), *FBXW7* (*n* = 12), *HRAS* (*n* = 10), *KRAS* (*n* = 10), and *CDKN2A* (*n* = 10) [15]. *TP53* was the most cited mutated gene among the studies reviewed. These genes were then used to identify potential targets in OPC-related HNC and to assess the feasibility of ongoing clinical trial strategies.

## 3. Results

### 3.1. Current Chemotherapy Strategies for the Treatment of HNC

According to the American Society of Clinical Oncology (ASCO) and the European Society for Medical Oncology (ESMO) clinical guidelines for HNC treatment [9], the most used chemotherapy drugs for HNC are cisplatin, 5-FU, cetuximab, docetaxel, and paclitaxel (Figure 2). Treatment combinations are proposed based on the status of the disease progression and contra-indications.

Cisplatin is an established chemotherapy used in the treatment of solid cancers, including HNC [16]. The compound crosslinks DNA, impeding DNA repair and inducing apoptosis [17]. Similarly, 5-fluorouracil (5-FU) inhibits DNA synthesis and is commonly combined with platinum chemotherapies, though it is less frequently used in antineoplastic therapies [18,19,20]. For refractory tumors, the epidermal growth factor receptor (EGFR) inhibitor cetuximab, and taxanes like docetaxel and paclitaxel, are currently accepted chemotherapies [21,22,23]. Combinations of these drugs have been found to improve patient survival outcomes compared to the use of a single treatment modality [19,23]. The combination of platinum chemotherapy, 5-FU, and cetuximab, known as the EXTREME regimen, aimed to improve overall survival (OS) [24]. The inclusion of cetuximab has shown slightly increased OS (median OS of 10.1 months) compared to platinum–fluorouracil alone (median OS of 7.4 months), at the expense of more toxicity which limited its widespread adoption. Currently, the Keynote 48 regimen is recommended as the standard first-line treatment for patients with recurrent or metastatic HNC [11,23,24]. However, for HPV-positive HNC, recent studies have cautioned the use of adjuvant cetuximab due to worse survival outcomes in comparison to standard platinum chemotherapy [25]. Inferior cetuximab responses in these patients further support the need for specific treatment recommendations based on HPV status.

Taxanes, such as paclitaxel and docetaxel, are semi-synthetic drugs that block the progression of the cell cycle [26,27,28]. Paclitaxel is an option for patients not eligible for platinum therapy [29]. The inclusion of taxanes in chemotherapy may reduce adverse side effects and the number of treatment cycles needed [24]. Studies that used taxanes have shown OS equal to or greater than 10.2 months [30], 14.7 months [23], and 21.3 months [31], and a progression-free survival (PFS) of 6.5 months [30], 5.2 months [23], and 5.8 months [31] compared to the EXTREME regimen which demonstrated an OS of 10.1 months and PFS of 5.6 months [32]. An exception to this increase in OS was observed in a study performed by Klinghammer et al. [33] that obtained a median OS of 8.9 months using docetaxel.

The inclusion of taxanes in chemotherapy provided alternatives to treat HNC patients [23,30]. The replacement of 5-FU with paclitaxel (100 mg/m^2^) was proposed in a phase II trial as first-line treatment in patients with HNC [34,35]. 5-FU causes adverse side effects, including oral mucositis and acute skin reactions, in addition to longer hospitalization for continuous intravenous infusion [34,35]. The switch to paclitaxel resulted in an overall response rate (ORR) of 40%, a median progression-free survival (PFS) of 5.2 months, and a median OS of 14.7 months [23]. Another phase II trial examined the CETMET regimen, consisting of cetuximab and paclitaxel/carboplatin, as a therapy for HNC. With a median PFS of 6.5 months, and a median OS of 10.2 months, this regimen had similar efficacy and less toxicity than standard treatment with cetuximab and 5-FU/cisplatin or carboplatin [36]. These findings coincide with a retrospective study evaluating a combination of cetuximab, paclitaxel, and carboplatin, which reported good tolerability and survival outcomes similar to the EXTREME regimen [8].

Similarly, docetaxel is another alternative in the EXTREME regimen [36]. Though one phase II trial found high toxicity rates, lower median OS, and no improvements in PFS with docetaxel in this regimen [33], a retrospective evaluation of biweekly treatment with docetaxel (50 mg/m^2^) and cetuximab (500 mg/m^2^) was shown to be safe and effective, with a median OS of 8.3 months and PFS of 4.0 months [31]. In platinum-resistant patients, a phase I/II trial revealed promising anti-tumor activity with docetaxel (75 mg/m^2^) and pembrolizumab (200 mg) followed by pembrolizumab maintenance therapy, with an ORR of 22.7%, median PFS of 5.8 months, and a median OS of 21.3 months [36]. These results are further supported by trials assessing regimens of docetaxel, cisplatin, and cetuximab, which reported high efficacies and patient survival, and favorable tolerability [22,24,37]. Therefore, treatment with cetuximab and docetaxel or paclitaxel is an alternative treatment strategy with satisfactory PFS and OS for cisplatin-resistant patients [29,31].

### 3.2. Immunotherapy for HNC Patients

Recently, immunotherapy has emerged at the forefront of anticancer therapy, with two PD-1 inhibitors, pembrolizumab and nivolumab, being approved for use in HNC (Figure 3) [38,39]. The CheckMate 141 trial compared treatment with nivolumab (3 mg per kilogram of body weight) to standard, single-agent, systemic therapy (methotrexate, docetaxel, or cetuximab) [40] Nivolumab treatment resulted in a longer OS and fewer toxicities compared to standard therapy [40]. Rischin et al. [10] demonstrated that the use of pembrolizumab as a monotherapy or in combination with chemotherapy, in addition to prolonging survival, maintains the quality of life of patients, and can be used as a first-line treatment for HNC [10].

The phase III clinical trial KEYNOTE-40 assessed the efficacy of standard treatment with methotrexate, docetaxel, or cetuximab with pembrolizumab in the second line or beyond setting [41]. Tumor PD-L1 expression predicted better outcomes for pembrolizumab, with a favorable safety profile and median OS of 8.4 months [41]. Similarly, the KEYNOTE-048 trial compared pembrolizumab in combination with chemotherapy (platinum and 5-FU) to cetuximab with chemotherapy as a first-line treatment [42]. The study reported an improved OS with immunotherapy, demonstrating that pembrolizumab in combination with chemotherapy is effective as a first-line treatment across all subgroups and pembrolizumab alone for patients whose tumors express PDL-1 in more than 1% of tumor cells [42]. During a 4-year follow-up period, both the first-line pembrolizumab and pembrolizumab associated with chemotherapy demonstrated a survival rate improvement in comparison to cetuximab with chemotherapy, with a subset of patients achieving a sustainable remission which was not previously possible with standard chemotherapy [42].

In all large phase III clinical studies, immunotherapy has been shown to be effective regardless of HPV status. Particularly, immunotherapies have emerged as potential de-escalated treatment options for HPV-positive HNC, with HPV-positive patients having some superior outcomes under these therapies. A study by Ferris et al. [40] reported a higher median OS following nivolumab treatment in patients with HPV-positive HNC, compared to those with HPV-negative tumors [40]. This corroborates the need for treatment individualization and de-escalation for HPV-associated HNC.

### 3.3. Current Status of Clinical Studies on HNC Treatments

Seventy-three clinical trials were found in HNC, 55 of which were related to OPC (Figure 4 and Table 1). Of these 55 clinical trials, 22 also included cancer in the oral cavity. The most common drugs used to treat HPV-related OPC are the same drugs used to treat HPV-negative HNC, with platinum and taxane-based systemic chemotherapies remaining the most common (Figure 4, Table 1). Indeed, platinum chemotherapy remains the most used intervention in the current clinical trials (*n* = 42), with cisplatin being the most common (*n* = 28), followed by carboplatin (*n* = 14) (Table 1). This is followed by taxane-based chemotherapies (*n* = 15), with paclitaxel being used more often in clinical trials (*n* = 12) than docetaxel (*n* = 3) (Table 1). Contrarily, other systemic chemotherapies like 5-fluorouracil (*n* = 4), gemcitabine hydrochloride (*n* = 1), hydroxyurea (*n* = 1), mitomycin C (*n* = 1), and 5-azacytidine (*n* = 1), are less widely used in the current clinical trials (Table 1). Similarly, targeted chemotherapies and novel immunomodulatory agents that remain experimental therapies in HNC, such as carbozantinib (*n* = 1), CUE-101 (*n* = 1), and TBio-6517 (n = 1), among others, are much less common among the current trials (Table 1, Figure 4). However, in HPV-positive HNC, immunotherapies are often incorporated into clinical trials (*n* = 15), with the PD-1 inhibitors nivolumab (*n* = 8) and pembrolizumab (*n* = 6) being the predominant immunotherapies studied in HNC (Table 1). Though they are slightly less common, targeted chemotherapies like cetuximab (*n* = 9), xevinapant (*n* = 1), vorinostat (*n* = 1), and bevacizumab (*n* = 1) are also implemented in some clinical studies (*n* = 12), with cetuximab being the most used (Table 1), likely due to its approval in this cancer type. In an attempt to reduce the side effects associated with radiation, many clinical trials involve a reduction in radiotherapy doses as a primary intervention (*n* = 5) or based on patient risk levels (*n* = 8), in combination with chemotherapy with or without immunotherapy (Table 1 and Figure 4).

The United States (*n* = 42; 77.7%) was the country with the highest number of ongoing clinical trials, followed by Canada (*n* = 3; 5.5%). These clinical trials began between the years 2000 and 2023 and will be completed between 2023 and 2032. The majority of the clinical trials (83.3%) commenced after 2015. Most studies (*n* = 44; 81.48%) considered HPV status as an inclusion criterion, while ten studies (18.52%) had no mention of HPV. Regarding the HPV status of participants, the p16 protein was the biological marker used to identify HPV positivity [43].

### 3.4. Novel Targets for Drug Treatments in HNC

Despite differences in patient outcomes based on HPV infections, the drugs approved and recommended for use in HNC remain the same regardless of HPV status. This leads to HPV-positive patients with superior prognoses undergoing unnecessarily intense treatments which can greatly reduce their quality of life. For this reason, strategies to de-intensify treatments in HPV-positive patients have been explored, including the use of targeted chemotherapy, immunotherapy, and neoadjuvant approach [44]. Due to the distinct genetic landscape and etiology of HPV-positive HNC, it is increasingly recognized as a unique HNC subtype, with its genetic and mutational profile offering new targets for de-escalated drug therapies. Recently, our group identified, based on public data repository, the genes that are commonly mutated in HPV-positive HNC, including *TP53*, *PIK3CA*, *PTEN*, *NOTCH1*, *RB1*, *FAT1*, *FBXW7*, *HRAS*, *KRAS*, and *CDKN2A* [15]. These genes were involved in proliferative and apoptotic mechanisms, supporting cancer cell growth when dysregulated [15]. As such, drugs that target these genes and their signaling pathways could provide alternative de-intensified treatment strategies for HPV-positive HNC (Figure 5).

Such drugs include cyclin-dependent kinase (CDK) 4 and 6 inhibitors, which may be effective in HNCs with mutated *RB1* and *CDK2NA*, that encode the tumor suppressors RB and p14/p16, respectively (Figure 5). The improper functioning of these regulators will lead to uncontrolled proliferation and cancer [45]. Though three CDK4/6 inhibitors are currently approved in the treatment of breast cancer, none have been approved for HNC despite early clinical and preclinical studies demonstrating their potential application [46]. Particularly, the CDK4/6 inhibitor palbociclib has shown promise when combined with cetuximab. When used as an adjuvant to radiotherapy in HNC, this combined therapy has shown promising efficacy and tolerability in a phase I study [47]. In patients with HPV-negative HNC, the addition of palbociclib to cetuximab induced moderate therapeutic responses, comparable to cetuximab with a placebo [48]. Further analyses have also shown improved survival with palbociclib in HNC patients with *CDKN2A* and *PI3KCA* mutations [49]. However, this inhibitor has not shown any clinical benefit in HPV-positive HNC, with palbociclib showing superior anti-tumor activity against HPV-negative HNC cells compared to HPV-positive cells in vitro [50]. The drug has also been found to induce significant adverse effects despite its efficacy in *CDKN2A*-mutated HNC [51]. Another phase 2 trial (NCT02101034) investigated palbociclib and cetuximab in cetuximab-resistant HPV-related OPC, but only one out of 24 patients achieved an objective response, suggesting that further investigation of this combination is not justified.

Other checkpoint inhibitors with potential applications in HNC are Wee1 inhibitors, which prevent cell cycle arrest [52]. Cancer cells with mutated tumor suppressors involved in the G1/S checkpoint, such as p53 [52] and RB [53], will rely on G2/M arrest for DNA repair, resulting in mitotic catastrophe and synthetic lethality when this mechanism is inhibited [52,54]. Though Wee1 inhibitors have shown promising anti-tumor effects in vitro and in vivo, few studies explore their clinical efficacy in HNC [52]. Nevertheless, in patients with HNC, the Wee1 inhibitor adavosertib has shown favorable safety, tumor response, and survival as an adjuvant to platinum chemoradiotherapy [55] as well as high tolerability and overall responses when combined with cisplatin and docetaxel [56].

Poly-adenosine diphosphate-ribose polymerase (PARP) inhibitors have also emerged as potential treatment alternatives in HNC, particularly among patients with mutations in *RB1* and *PTEN*. These genes encode tumor suppressors that play a role in cell cycle checkpoints and DNA repair [54,57]. As PARP is also involved in the repair of DNA damage, its inhibition in *RB1*- or *PTEN*-deficient cancer cells can lead to genomic instability and cell death [58,59,60]. Despite promising preclinical results [61], PARP inhibitors have not been widely accepted in clinical settings for the treatment of HNC. However, they have resulted in favorable outcomes as adjuvants in preliminary clinical studies [61]. When combined with radiotherapy or chemoradiotherapy for HNC, the PARP inhibitor olaparib has been found to maintain high survival outcomes with good safety and tolerability [62,63]. Similarly, the combination of olaparib with carboplatin and the PD-1 inhibitor pembrolizumab was found to induce an overall response rate of 67% when administered as a primary treatment for recurrent or metastatic HNC. As such, PARP inhibitors may provide alternative treatments for HNC, though further investigations are needed for their implementation.

*BRAF* and *MEK* inhibitors are commonly used in combination treatments for other cancers with high rates of *RAS* and *CDKN2A* mutations [61,64,65]. With the success of these inhibitors in other cancers, their application in HNC could provide new treatment avenues. Though the effects of these drugs in HNC remains unknown, two studies have shown that the MEK inhibitor trametinib may be beneficial against HNC in vitro. The inhibitor has been shown to improve the efficacy of PD-1/PD-L1 immunotherapies in HNC cells [66] and to induce partial cell death in HNC cell lines [67]. Further investigations are required to understand the implications of *BRAF* and *MEK* as potential targets in HNC.

Similarly, the farnesyltransferase inhibitor tipifarnib may be a possible alternative treatment for HNC patients with *RAS* mutations, particularly *HRAS* mutations (Figure 5). Though larger studies are underway (NCT03719690), the drug has shown promising responses and survival rates in HNC patients with *HRAS* mutations [68].

HSP90 inhibitors may also be possible treatment options in HNC, particularly in tumors with mutated *TP53* genes [69]. As certain cancers rely on the HSP90-mediated stabilization of mutated p53, these inhibitors have been shown to promote cancer cell death and improve survival in vivo [70]. HSP90 inhibitors have also been shown to reduce the expression of pro-proliferative agents, including mutated p53, and promote the production of p21, which halts the cell cycle in vitro [71]. Despite these findings, research on HSP90 inhibitors in HNC is limited, with only one study reporting their benefit in enhancing platinum chemoradiotherapy in HNC cell lines [72].

Finally, AKT and mTOR inhibitors may also be beneficial in HNC treatments, particularly for patients with mutations in *PIK3CA*, *PTEN*, and *RAS* genes. Indeed, the inhibition of downstream effectors of the anti-apoptotic PI3K/AKT pathway has shown success in other cancer types with these mutations [73,74] and has shown promise for HNC treatment in preclinical studies [75]. Few clinical trials have assessed the efficacy of AKT inhibitors in HNC, and mostly reported unfavorable tumor responses [75,76,77]. However, a recent meta-analysis has highlighted the potential benefits of mTOR inhibitors as adjuvants in combination therapies for HNC, though further research is required as these drugs did not induce significant tumor responses when administered alone [78].

## 4. Discussion

Regardless of the patient’s HPV status, cisplatin, nivolumab, cetuximab, paclitaxel, pembrolizumab, 5-fluorouracil, and docetaxel are the drugs currently used to treat HNC. Due to the high toxicity of most available treatment options, the search for a therapeutic drug presenting high specificity, efficacy, tolerability, and reduced side effects is under investigation and the subject of ongoing clinical trials. Several factors are involved for the selection of the ideal therapy, including the balance between the impact on the patient’s quality of life and the real clinical benefits regarding the survival rates [79].

HPV-associated HNC has clinical and molecular behaviors that differ from HPV-negative tumors [22,37]. Indeed, despite both HNC subtypes having similar differential gene expression, superior prognostic outcomes are seen in HPV-positive HNC [6]. For this reason, novel de-escalated strategies should be clinically trialed in HPV-positive patients to reduce adverse effects. These include advances in adjuvant and neoadjuvant chemotherapies and immunotherapies, a reduction in treatment intensity, or the incorporation of new drugs. Taxanes have shown promise in the treatment of HNC, inducing favorable outcomes when integrated into combination therapies, and providing alternatives for platinum-refractory tumors [21,22,23]. Similarly, the EGFR inhibitor cetuximab has been successfully applied in HNC treatments [32], though strategies involving the drug have been under debate for HPV-positive patients [25]. Conversely, PD-1/PD-L1 inhibitors like nivolumab and pembrolizumab have shown great efficacy against HNC, particularly in HPV-positive patients. Although these drugs have been approved for use in HNC, their success may translate into de-intensified strategies for HPV-positive patients. With a small number of clinical trials in favor of individualizing treatment for HPV-associated HNC, further investigations are needed to broaden treatment options in these patients and to reach a conclusion. Clinical trials focusing on experimental drugs are critical in bridging current treatment gaps and improving patient prognosis and quality of life.

Currently, clinical data that support the use of alternative drugs for the treatment of the 10 most mutated genes (*TP53*, *PIK3CA*, *PTEN*, *NOTCH1*, *RB1*, *FAT1*, *FBXW7*, *HRAS*, *KRAS*, and *CDKN2A* [15]) in HPV-positive HNC are limited. Despite showing potential as novel HNC targets, few registered ongoing clinical trials are investigating drugs that may target these genes (Table 1). Potential treatment alternatives include BRAF/MEK inhibitors, which have been useful in the treatment of various cancers like CDKN2A-negative melanoma, and CDK4/6 inhibitors, which have been approved to treat breast cancer. Apart from FAT1, the 10 commonly mutated genes mentioned above are druggable, warranting the exploration of potential targeted therapies. However, the successes of other novel therapeutic strategies in HNC cannot be denied. Xevinapant, a pro-apoptotic agent, has shown promising improvements in reducing mortality risks and sustaining locoregional control, though the latter was non-significant at 3-year follow-up [80,81]. Similarly, alternative chemotherapy dosing regimens, like metronomic chemotherapy, have been found to improve progression-free and overall survival [82,83]. Nevertheless, these strategies are not yet validated in the clinical context of HNC, as research on this cancer type is limited, especially for HPV-associated OPC. Thus, further research is required to elucidate the implications of new gene targets and therapeutic regimens in HNC.

## 5. Conclusions

Despite ongoing efforts to enhance the prognosis of patients with HNC through diverse therapeutic approaches, the complexity and heterogeneity of the disease have delayed the desired revolutionary advancements in treatment. Numerous medications have undergone investigation in clinical trials, either as a single treatment or in combination with other drugs. The effectiveness of chemotherapy, chemoradiotherapy, targeted therapy, and immunotherapy in treating HNC varies based on factors such as disease stage, comorbidities, age, and prior treatments. Immunotherapy has gained prominence due to the pivotal role of the immune system in HNC carcinogenesis. While targeted therapeutics capitalize on molecular insights into cancer biology, their response is limited by the intricate interplay of multiple cell-signaling pathways. To establish their efficacy across diverse HNC cohorts and stages, additional clinical trials that involve innovative approaches and/or targets are necessary.

## Figures and Tables

**Figure 1 ijms-25-04009-f001:**
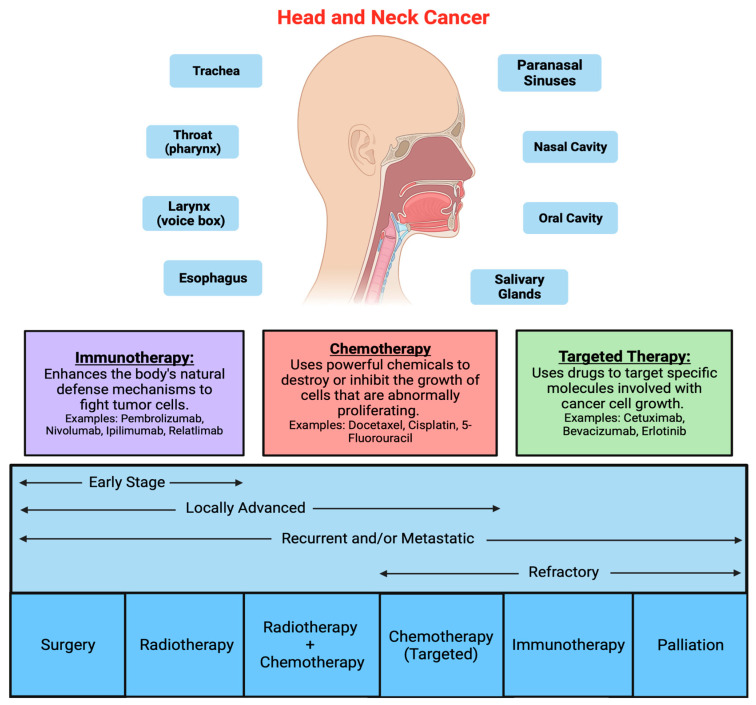
Head and neck cancer (HNC) involves a heterogeneous group of malignant tumors that can affect different sites of the oral cavity, pharynx, and larynx and upper respiratory tract. Treatment plans are based on the clinical and pathological stage of the cancer and consist of surgery, radiation therapy, chemotherapy (red box), immunotherapy (purple box), target therapy (green box), or a combination of these treatments’ modalities.

**Figure 2 ijms-25-04009-f002:**
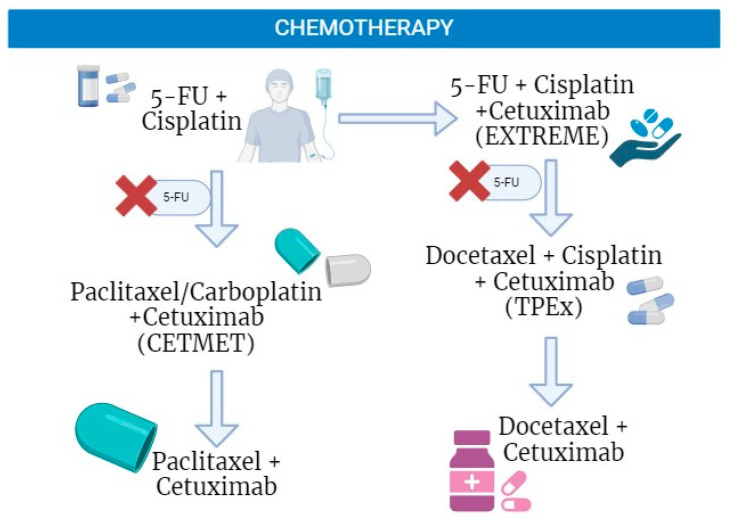
Current chemotherapy regimens used in the treatment of head and neck cancer. Many clinical trials have studied the benefits of combination therapy for the treatment of head and neck cancer. These studies have concluded that combinations of cetuximab, taxanes, and cisplatin are beneficial treatment alternatives to the EXTREME regimen. This figure was generated using biorender.com.

**Figure 3 ijms-25-04009-f003:**
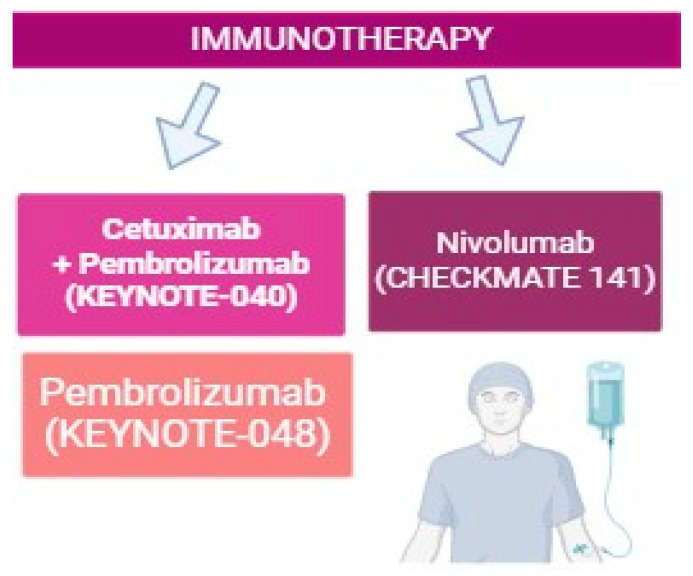
Current immunotherapy regimens used in the treatment of head and neck cancer. Three landmark clinical studies, the Keynote-040, Keynote-048, and Checkmate 141 trials, were crucial in the approval of PD-1/PD-L1 immunotherapies in HNC. This figure was generated using biorender.com.

**Figure 4 ijms-25-04009-f004:**
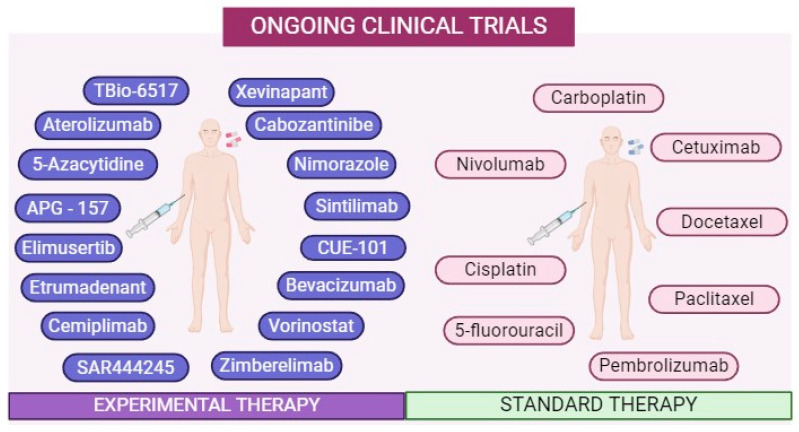
Summary of the drugs in ongoing clinical trials registered in public platforms (https://clinicaltrials.gov). The panel on the right represents the most common drugs used to treat both HPV-positive and HPV-negative oropharyngeal cancer. The panel on the left represents the novel therapeutics currently in phase I, phase II, and phase III clinical trials. This image was created using biorender.com.

**Figure 5 ijms-25-04009-f005:**
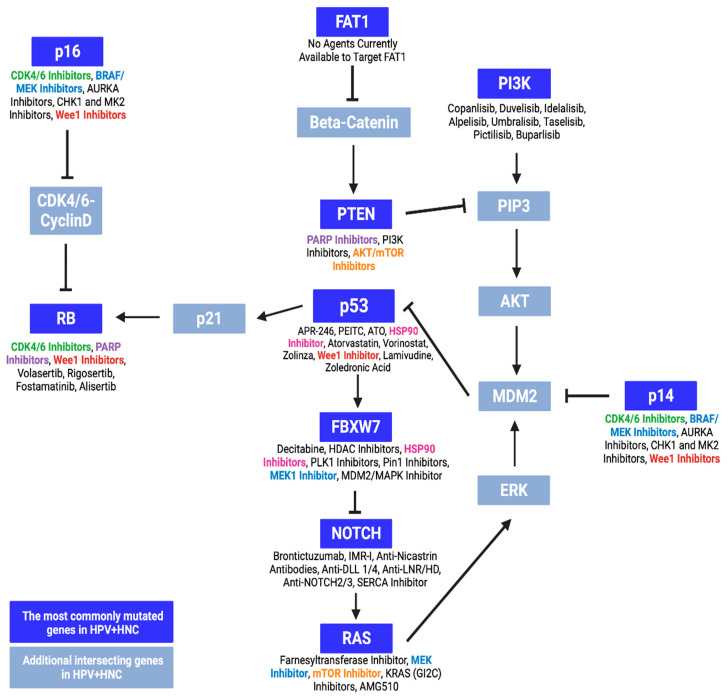
Proteins expressed by the most commonly mutated genes in HPV-associated head and neck cancer (HNC) and their intersecting pathways. These mutated genes include *TP53*, *PIK3CA*, *PTEN*, *NOTCH1*, *RB1*, *FAT1*, *FBXW7*, *HRAS*, *KRAS*, and *CDKN2A*. The genes and proteins in blue are intersecting genes in the HPV-associated HNC pathway. The drugs currently available, or in development, to target these mutations are listed below with their respective genes/proteins. Some common drugs include HSP90 inhibitors, BRAF/MEK inhibitors, CDK4/6 inhibitors, PARP inhibitors, mTOR inhibitors, and Wee1 inhibitors. Different colored font was added to indicate that certain treatments can target multiple genes and pathways.

**Table 1 ijms-25-04009-t001:** Ongoing clinical trials using chemotherapies, targeted therapies, and immunotherapies to treat OPC.

NCT Number	Study Status	HPV Status	Interventions	Phase	Sample Size	Start and Completion Date	Country
NCT05724602	NR	+	Debio 1143, PLCB	2	230	09/2023–10/2029	NP
NCT05721755	NYR	+	CBDCA, CDDP, 5-FU, PTX, MK-3475	3	290	04/2023–03/2030	NP
NCT05608369	NYR	−	CDDP, SAHA	2	64	05/2023–02/2024	US
NCT05541016	NYR	+	CDDP, TXT	2	320	02/2023–08/2029	US
NCT05535023	NYR	+	SAR444245, REGN2810	2	26	02/2023–10/2024	US
NCT05419089	R	+	RDR, CDDP	2	199	07/2022–06/2027	US
NCT05317000	NYR	+	5-AC, NIVO	1	50	02/2023–02/2026	NP
NCT05312710	R	NM	APG-157	2	24	04/2022–04/2023	US
NCT05268614	R	+	RT or RDR, CDDP or CBDCA and PTX	2	250	05/2022–06/2032	US
NCT05136196	R	+	Cabozantinib S- malate, NIVO	2	150	10/2022–10/2025	US
NCT05108870	R	+	CBDCA, PTX	1, 2	98	08/2022–01/2026	US
NCT05063552	R	+	MPDL328OA, rhuMab-VEGF, CBDCA, C225, CDDP, TXT	2, 3	430	12/2021–12/2027	US
NCT04900623	R	+	RT or RDR, CDDP or CBDCA and PTX	2	75	07/2021–06/2032	US
NCT04892875	NYR	+/−	AB122, AB 928, CDDP	1	24	02/2023–04/2025	US
NCT04862650	R	+	CBDCA, REGN2810, PTX	2	42	11/2021–12/2024	US
NCT04852328	R	+	CUE-101	2	30	12/2021–10/2025	US
NCT04718415	R	NM	IBI308, PTX, CBDCA	2	25	01/2021–05/2026	CN
NCT04576091	R	+	BAY-1895344, MK-3475	1	37	02/2021–04/2026	US
NCT04572100	R	+	PTX, CBDCA RT or RDR	1	36	10/2020–03/2023	US
NCT04502407	R	+	RT, CDDP	2	36	02/2021–09/2025	US
NCT04444869	R	+	CDDP	2	28	09/2020–06/2025	US
NCT04301011	ANR	+	MK-3475	1,2	27	06/2020–12/ 2023	US, CA, KR
NCT04180215	R	+	HB-201, HB-202	1, 2	200	12/2019–06/2025	US
NCT04124198	R	NM	CDDP, Nimorazole	-	138	03/2019–01/2029	DK
NCT04106362	R	+	C225, CDDP	2	70	01/2020–07/2024	US
NCT03829722	ANR	+	NIVO, CBDCA, PTX	2	26	09/2019–09/2024	US
NCT03822897	ANR	+	RT ± CDDP	2	103	02/2019–12/2024	CA
NCT03799445	R	+	MDX-CTLA-4, NIVO	2	180	07/2019–12/2023	US
NCT03715946	ANR	+	NIVO, RDR	2	42	11/2018–11/2023	US
NCT03646461	ANR	E	PCI-32765, C225, NIVO	2	5	10/2018–05/2024	US
NCT03621696	ANR	+	CDDP, RT	2	63	10/2018–03/2026	US
NCT03410615	ANR	+	CDDP, MEDI4736, CP-675	2	129	01/2018–07/2026	BE, CA
NCT03383094	R	+	MK-3475, CDDP	2	114	03/2018–06/2024	US
NCT03370276	ANR	+	NIVO, C225	1, 2	95	12/2017–11/2023	US
NCT03323463	R	+	RDR, CDDP, CBDCA, 5-FU	2	300	10/2017–10/2024	US
NCT03258554	ANR	+	C225, MEDI4736	2, 3	493	12/2017–12/2025	US
NCT03215719	R	+	RT or RDR, CDDP	2	54	07/2017–12/2025	US
NCT03174275	ANR	-	MEDI4736, CBDCA, nab-PTX, CDDP	2	39	12/2017–12/ 2026	US
NCT03107182	ANR	+	nab-PTX, CBDCA, NIVO, CDDP, HU, 5-FU	2	76	06/2017–07/2023	US
NCT03088059	R	-	BIBW 2992 MA2, PD-332991, IPH2201, MEDI4736, CJNJ-6765200, INCAGN01876	2	340	11/2017–12/2025	BE
NCT03082534	ANR	NM	MK-3475, C225	2	78	03/2017–05/2024	US
NCT03077243	ANR	+	RT or RDR, CDDP	2	215	12/2016–02/2026	US
NCT02918955	R	NM	RT or RDR, CDDP	3	65	10/2016–03/2030	CH
NCT02586207	ANR	NM	CDDP, MK-3475	1	59	11/2015–09/2023	US
NCT02573493	ANR	+	nab-PTX, CDDP, C225	2	96	04/2016–12/2029	US
NCT02369458	ANR	+	MTC, HSP-130	2	48	04/2015–06/2023	US
NCT02281955	ANR	+	RT or RDR, CDDP	2	115	08/2014–11/2024	US
NCT02254278	ANR	+	CDDP, RDR	2	316	10/2014–05/2024	US
NCT02229656	ANR	−	RT, AZD2281	1	12	09/2014–01/2024	NL
NCT01855451	ANR	+	C225, RT, CDDP	3	189	06/2013–08/2023	AU
NCT01706939	ANR	+	RDR, CBDCA	3	23	09/2012–05/2035	US
NCT00956007	ANR	NM	C225, RT	3	703	11/2009–08/2029	US
NCT00544414	ANR	E	CDDP, TXT, 5-FU, dFdCyd, leucovorin	2	30	06/2000–12/2023	NP
NCT00494182	ANR	NM	CBDCA, PTX, Sorafenib	2	48	04/2007–05/2023	US
NCT03719690	ANR	NM	Tipifarnib	2	284	11/2018–05/2023	US

Abbreviations: NM: Not mentioned; NR: Not recruiting; ANR: Active, not recruiting; R: Recruiting; NYR: Not yet recruiting; NP: Not provided; E: Evaluate; US: United States; CN: China; AU: Australia; NL: Netherlands; BE: Belgium; CA: Canada; DK: Denmark; CH: Switzerland; KR: Korea. RT: Standard Radiotherapy; RDR: Reduced Dose Radiation; PTX: paclitaxel; 5-FU: 5-Fluorouracil; CDDP: Cisplatin; C225: Cetuximab; TXT: Docetaxel; Debio 1143: Xevinapant; MK-3475: Pembrolizumab; PLCB: Placebo; CBDCA: Carboplatin; SAHA: Vorinostat; dFdCyd: Gemcitabine hydrochloride; HU: Hydroxyurea; NIVO: Nivolumab; MTC: Mitomycin-C; MDX-CTLA-4: Ipilimumab; 5-AC: 5-azacytidine; rhuMab-VEGF: Bevacizumab.

## Data Availability

The datasets used and/or analyzed during the current study are available from the corresponding author upon reasonable request.

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
