# Peer review of "Therapeutic Advances and Challenges for the Management of HPV-Associated Oropharyngeal Cancer"

_ijms, 2024, doi:10.3390/ijms25074009_

Round 1
Reviewer 1 Report
Comments and Suggestions for Authors
In the manuscript by Isis de Araújo Ferreira Muniz et al the authors summarize their review efforts on the topic of treatments in head and neck cancer. The review is informative and easy to read and mostly well written.
On the other hand there appears to be a mismatch between the main topic (new treatments in ongoing studies – excluding completed studies in review) and the manuscript text that puts a large emphasis on established treatments and the discussion thereof. Better connection of the clinical trial review and current literature review parts of the manuscript is warranted. The discussion about most mutated genes is also helpful but should be supported by additional literature instead only by authors previous study. Without going into details, it is really eyebrow raising to see p53 as the most prominent (first) gene in the list of genes mutated in HPV-associated HNSCC. There are some minor formatting issues to polish but no major editing or typo errors are notable.
Detailed comments are listed below in page line format:
P3 L112 consider rounding percentages to 1 decimal place.
P3 L114 the text implies the dose reduced radiation therapy was employed, however, this information about the dose is not shown in accompanying table 1 (ie „RT ↓“ symbol could be used to indicate dose reduced RT). Do studies on HPV associated cases really have dose reduced RT at this time?
P4 Table 1. the row for NCT03174275 is split across 2 pages. Consider moving both lines of the same row to the same page. The abbreviation „=/-„ for HPV status in study NCT02369458 is not explained. Control arms should be indicated somehow for each trial (ie always at the end of the list preceded by vs?)
P4/5 the study apparently put a focus on currently running clinical trials in an effort to present new state of the treatment landscape. However, within the text, there is comparably lower amount of content regarding the new agents while there is a lot of text about the established treatments. At least the new compounds shown on Figure 2 should be accompanied by the number of studies the compounds are found in (unless presenting Sintilimab two times is the only indication of more than one study looking into a single new agent?). For established treatments some targets are shown, but no information is shown where do the new agents fit in the treatment landscape
P5 L132 Figure 2 suggests that different colors were used for different phases of clinical trials, but all experimental therapies use the same color for labels.
P6 L141 last sentence of the paragraph got separated in style from the rest of the paragraph.
P6 L143 resolution of Figure 3 is suboptimal in the PDF provided. The arrows imply some time or logical flow which is probably not intended. Ie top-leftmost patient is initially undergoing 5-FU+cispt treatment and then goes to EXTREME protocol?
P7 L170-188 it might be interesting to make a figure or table summarizing the different protocols and the OS/PFS metrics that are reported in the text.
P8 L232 the section about novel targets should be combined/crossreferenced with the Table1/Figure 2. Do any of the newly investigated agents fall under paragraphs describing CDK (L252), CKI (L270), PARP (L279), BRAF/MEK (L294), HSP (L307), AKT/mTOR (L315) inhibitors?
P8L233 paragraph formatting is different.
P8L240 it is unusual to see p53 on the first place as commonly mutated in HPV associated HNSCC since HPV E6 readily deactivates p53 and its mutation status is irrelevant. It would be beneficial to show the percentages of mutations in the listed genes for HPV associated cancers based on the authors analysis of 8000+ cases and sort them in descending order. While it is understandable that the authors prefer to cite their previous work, the list of genes should still be put into context of the seminal analysis by the TCGA consortium https://www.nature.com/articles/nature14129
P9 L244 Figure 4. While very interesting, the figure might be further improved by linking the content with the novel therapies listed in Table1/Figure 2. Is there some overlap between Figure 2 and 4? The color coding legend for inhibitors is missing.
P9 L251 some separation of figure caption and the rest of the text is warranted
P10 L350 it is unclear why the study NCT03719690 is discussed as ongoing yet it was not shown in table 1? But showing the studies listed in table 1 in the same way as NCT03719690 is shown in the text would be helpful to put the ongoing trials in context.
Author Response
In the manuscript by Isis de Araújo Ferreira Muniz et al the authors summarize their review efforts on the topic of treatments in head and neck cancer. The review is informative and easy to read and mostly well written.
On the other hand there appears to be a mismatch between the main topic (new treatments in ongoing studies – excluding completed studies in review) and the manuscript text that puts a large emphasis on established treatments and the discussion thereof. Better connection of the clinical trial review and current literature review parts of the manuscript is warranted. The discussion about most mutated genes is also helpful but should be supported by additional literature instead only by authors previous study. Without going into details, it is really eyebrow raising to see p53 as the most prominent (first) gene in the list of genes mutated in HPV-associated HNSCC. There are some minor formatting issues to polish but no major editing or typo errors are notable.
Detailed comments are listed below in page line format:
P3 L112 consider rounding percentages to 1 decimal place.
Response: The percentages were removed altogether for clarity (page 3, lines 112-113), as rates of drug use in clinical trials were found to vary in the literature reviewed.
P3 L114 the text implies the dose reduced radiation therapy was employed, however, this information about the dose is not shown in accompanying table 1 (ie „RT ↓“symbol could be used to indicate dose reduced RT). Do studies on HPV associated cases really have dose reduced RT at this time?
Response: Reduced-dose radiation has become a common treatment de-escalation strategy used in clinical trials for HPV-positive HNC. The “Interventions” column in Table 1 was modified to specify the studies that use reduced-dose RT (pages 4-5). Line 116 (page 3) was modified to indicate that studies assessing reduced-dose RT can be found in Table 1.
P4 Table 1. the row for NCT03174275 is split across 2 pages. Consider moving both lines of the same row to the same page. The abbreviation „=/-„ for HPV status in study NCT02369458 is not explained. Control arms should be indicated somehow for each trial (ie always at the end of the list preceded by vs?)
Response: Table 1 was modified so NCT03174275 is on one page (page 5, Table 1), and the notation “=/-” was clarified (page 4, Table 1). The majority of the clinical trials do not provide the information regarding the control arms, so, the descriptions of the interventions was mentioned in the “Interventions” column of Table 1 (page 4-5).
P4/5 the study apparently put a focus on currently running clinical trials in an effort to present new state of the treatment landscape. However, within the text, there is comparably lower amount of content regarding the new agents while there is a lot of text about the established treatments. At least the new compounds shown on Figure 2 should be accompanied by the number of studies the compounds are found in (unless presenting Sintilimab two times is the only indication of more than one study looking into a single new agent?). For established treatments some targets are shown, but no information is shown where do the new agents fit in the treatment landscape
Response: We reformatted the paper to improve the flow of the text and make it clear the paper is exploring existing approved therapies, treatments currently under investigation, and lastly potential treatment avenues that can be further explored in the context of head and neck cancer.
P5 L132 Figure 2 suggests that different colors were used for different phases of clinical trials, but all experimental therapies use the same color for labels.
Response: The colored labels indicating the clinical trial phase were removed for clarity in Figure 2 (page 5, line 133).
P6 L14 last sentence of the paragraph got separated in style from the rest of the paragraph.
Response: The formatting issue was resolved, and the entire sentence is in the same style (page 6, lines 142-143).
P6 L143 resolution of Figure 3 is suboptimal in the PDF provided. The arrows imply some time or logical flow which is probably not intended. Ie top-leftmost patient is initially undergoing 5-FU+cispt treatment and then goes to EXTREME protocol?
Response: We have separated Figure 3 into two figures, one for chemotherapy and one for immunotherapy, to avoid confusion and improve logical flow. We have also reformatted the figures to provide a logical flow of the major clinical trials that were published. The quality of the images were also improved (pages 2,4, 6,10,11).
P7 L170-188 it might be interesting to make a figure or table summarizing the different protocols and the OS/PFS metrics that are reported in the text.
Response: We have reached the maximum number of figures allowed for our manuscript, with a total of 5 figures and 1 large table. To accommodate this constraint, we have opted to include survival data within the main text rather than adding an additional table. Additionally, the chemotherapy regimens are comprehensively described in Figure 3 (page 6, line 146).
P8 L232 the section about novel targets should be combined/crossreferenced with the Table1/Figure 2. Do any of the newly investigated agents fall under paragraphs describing CDK (L252), CKI (L270), PARP (L279), BRAF/MEK (L294), HSP (L307), AKT/mTOR (L315) inhibitors?
Response: Based on the ongoing studies listed in Table 1 (pages 4-5) and the treatments mentioned in Figure 2 (page 5, line 134), there is no overlap with the new agents that may target the identified genes. This has been emphasized in the discussion (page 11, lines 365-367).
P8L233 paragraph formatting is different.
Response: The formatting of this paragraph was fixed (page 8, lines 230-235).
P8L240 it is unusual to see p53 on the first place as commonly mutated in HPV associated HNSCC since HPV E6 readily deactivates p53 and its mutation status is irrelevant. It would be beneficial to show the percentages of mutations in the listed genes for HPV associated cancers based on the authors analysis of 8000+ cases and sort them in descending order. While it is understandable that the authors prefer to cite their previous work, the list of genes should still be put into context of the seminal analysis by the TCGA consortium https://www.nature.com/articles/nature14129
Response: In the paper published (Atique, M.; Muniz, I.; Farshadi, F.; Hier, M.; Mlynarek, A.; Macarella, M.; Maschietto, M.; Nicolau, B.; Alaoui-Jamali, M.A.; da Silva, S.D. Genetic Mutations Associated with Inflammatory Response Caused by HPV Integration in Oropharyngeal Squamous Cell Carcinoma. Biomedicines 2024, 12, 24. https://doi.org/10.3390/biomedicines12010024) we showed a supplementary table and a list of all mutated genes with highlighting on the most cited genes, which are TP53 (n = 22), PIK3CA (n = 20), PTEN (n = 16), NOTCH1 (n = 14), RB1 (n = 13), FAT1 (n = 13), FBXW7 (n = 12), HRAS (n = 10), KRAS (n = 10), and CDKN2A (n = 10). We added this information in the paper. In the same published paper, the TCGA was used to validate the alterations. Please note the image below, based on TCGA information, we created a lollipop graph to show that the chromosome region associated with mutation in TP53 in not identical for HPV positive and HPV negative head and neck cancer. The team is investigating this information in a large study within our research institution.
(image in the file attached for the reviewer): Lollipop graph showing TP53 mutations associated with HPV positive (top) and HPV negative (bottom).
P9 L244 Figure 4. While very interesting, the figure might be further improved by linking the content with the novel therapies listed in Table1/Figure 2. Is there some overlap between Figure 2 and 4? The color-coding legend for inhibitors is missing.
Response: There is not much overlap between the figures since Figure 2 described drugs that are part of ongoing clinical trials, whereas Figure 4 described current treatments that are commonly used to treat many types of cancer. These common cancer treatments are then placed in association with the HPV-related head and neck cancer genes that they could potentially target. Please note that Figure 2 rearranged in this revision to avoid confusion.
P9 L251 some separation of figure caption and the rest of the text is warranted
Response: A space was added between the legend of Figure 5 and the text (page 9, lines 264-265).
P10 L350 it is unclear why the study NCT03719690 is discussed as ongoing yet it was not shown in table 1? But showing the studies listed in table 1 in the same way as NCT03719690 is shown in the text would be helpful to put the ongoing trials in context.
Response: The ongoing clinical trials listed in Table 1 is not discussed in depth, as they do not yet have results to report. Rather, Table 1 provides information on the current status of clinical trials for HNC and the types of therapies under investigation, while sections 3.2 (pages 6-7) and 3.3 (pages 7-8) of the text provide information on studies that supported the approval of current treatment regimens. For this reason, the NCT numbers of ongoing studies are not present in-text. NCT03719690 was included in table 1 (page 9).

Reviewer 2 Report
Comments and Suggestions for Authors
Due to still not satisfying the survival time of patients, oropharyngeal cancer/head and neck cancer is a severe problem for clinicians. As the authors mentioned, HPV status is an essential factor influencing the effectiveness of anti-cancer treatment. The manuscript presents current clinical studies in the field of oropharyngeal cancer treatment, chemotherapy, and immunotherapy possibilities, as well as targets for novel drugs based on commonly mutated genes in HPV-positive tumors.
The knowledge implemented in this article will be helpful for readers interested in head and neck cancer biology and treatment. In my opinion, several editorial improvements can help to make the manuscript more clear.
1. Table 1, the symbol "*" below the table can be replaced with the word "Abbreviations:"; in the table "NM" abbreviation was used, and in the legend "NT" - please unify; what does it mean "=/-" in Status HPV?
2. In Figure 2, the name "Sintilimab" is doubled.
3. I suggest diving the left and right parts of Figure 3 into new Figure 3 and Figure 4. Then, Fig. 3 should be placed at the end of paragraph 3.2. and Fig. 4 at the end of paragraph 3.3. Moreover, the cartunes used in current Fig. 3 have no meaning and seem to be located randomly. Please, prepare corrections of this schemes that will be better to follow by the reader and more accessible to follow based on the main text. I also think that "chemotherapy" does not reflect cetuximab (targeted therapy), so targeted therapy should also be mentioned in this Fig.
4. Please include the full name of all abbreviations when first time mentioned, e.g., OS or PFS.
5. The authors highlighted that "HPV-associated HNC has clinical and molecular behaviors that differ from HPV-negative tumors". There is a review article (10.3390/cancers15174247) discussing the same point of view, focusing primarily on HPV-negative tumor treatment. It can be potentially used in this part of the Discussion.
6. In lines 349 and 350 the gene names should be written in italics.
Author Response
Due to still not satisfying the survival time of patients, oropharyngeal cancer/head and neck cancer is a severe problem for clinicians. As the authors mentioned, HPV status is an essential factor influencing the effectiveness of anti-cancer treatment. The manuscript presents current clinical studies in the field of oropharyngeal cancer treatment, chemotherapy, and immunotherapy possibilities, as well as targets for novel drugs based on commonly mutated genes in HPV-positive tumors. The knowledge implemented in this article will be helpful for readers interested in head and neck cancer biology and treatment. In my opinion, several editorial improvements can help to make the manuscript more clear.
1. Table 1, the symbol "*" below the table can be replaced with the word "Abbreviations:"; in the table "NM" abbreviation was used, and in the legend "NT" - please unify; what does it mean "=/-" in Status HPV?
Response: Table 1 was modified to address these comments. The word “Abbreviations” was added in place of “*” (page 5, line 125), the abbreviation “NM” was unified with the legend (page 5, line 125), and the “=/-” symbol was clarified.
2. In Figure 2, the name "Sintilimab" is doubled.
Response: Figure 2 was edited to only include one “Sintilimab”.
3. I suggest dividing the left and right parts of Figure 3 into new Figure 3 and Figure 4. Then, Fig. 3 should be placed at the end of paragraph 3.2. and Fig. 4 at the end of paragraph 3.3. Moreover, the cartunes used in current Fig. 3 have no meaning and seem to be located randomly. Please, prepare corrections of this schemes that will be better to follow by the reader and more accessible to follow based on the main text.
Response: We agreed with the reviewer. Figure 3 was divided in two as recommended and the explanations followed in the main text.
4. Please include the full name of all abbreviations when first time mentioned, e.g., OS or PFS.
Response: The full names for overall survival (OS) and progression-free survival (PFS) were added on page 6 line 160, and page 7 line 173, respectively.
5. The authors highlighted that "HPV-associated HNC has clinical and molecular behaviors that differ from HPV-negative tumors". There is a review article (10.3390/cancers15174247) discussing the same point of view, focusing primarily on HPV-negative tumor treatment. It can be potentially used in this part of the Discussion.
Response: The Discussion was modified to highlight key prognostic differences between HPV-positive and –negative HNC patients, which justifies the use of de-escalated treatment strategies in HPV-positive HNC (page 11, lines 334-340). The review article mentioned (Kleszcz 2023) was cited to support the added information (page 11, lines 334-335).
6. In lines 349 and 350 the gene names should be written in italics.
Response: The gene names were italicised on page 11, lines 350-351.
Round 2
Reviewer 1 Report
Comments and Suggestions for Authors
In the revised manuscript by de Araújo Ferreira Muniz et al. the authors addressed most of the more technical issues with the original.
However, some issues remain or were introduced by the revision. Unfortunately, there is no clean text version of the manuscript so tracking comments is nearly impossible and the response to reviewers letter doesn’t cross reference to the revised manuscript at all.
Even if ignoring the technical aspects of the revised manuscript the critical content issues are significant
1) Comment about dose reduced RT (previously at P3L114) remains partially addressed. Indeed, in the revised table 1 now on page 5(?) the authors introduced the RDR and LDR abbreviations as dose reduced and low dose radio therapy. Ie for study NCT01706939. However, for study NCT02254278 the therapy states „Reduced-dose RT“ in full.
More importantly, of 55 currently ongoing studies, only 1 is listed as LDL (NCT04572100) and only 2 are listed as RDR (NCT01706939, NCT02254278). On the other hand, 15 still included only RT (suggestive of not reduced RT).
In light of this, is the following claim still factual: „Most clinical trials involved a reduction in the dose of radiation in combination with chemotherapy, … in an attempt to reduce the side effects (Table 1, Figure 2).“?
2) The original comment dealing with pages 4 and 5 was partially addressed as well. The authors state “We reformatted the paper to improve the flow of the text and make it clear the paper is exploring existing approved therapies, treatments currently under investigation, and lastly potential treatment avenues that can be further explored in the context of head and neck cancer.”
The manuscript indeed has a lot of marked changes, however, the actual order of topics is unchanged from the initial document. At the start of the results (P3L113) the current ongoing studies are presented, followed by lengthy presentation of current treatments and finished by mutations. Critically, no summary of Table 1 is provided. Figure 2 still does not indicate how many studies actually investigated novel agents (eve if it is one each) or novel combinations (but at least Sintillimab is not erroneously repeated as before). There are 55 studies but only 16 experimental and 8 standard agents listed meaning that a lot of content is not summarized. Please don’t ask readers to browse through Table 1 spanning multiple pages and count how many occurrences of CCDP, TXXT, CBDCA,PTX …etc are there to have a grasp on what is investigated more often and what is not investigated often at this time.
Still no textual connection between different aspects of the work is present. No new text was added, only the paragraphs marked as revised.
3) Page 3 L114 “54 of which were related to OPC (Figure 2; Table 1)” is erroneous since Table 1 now contains 55 trials but the sentence was not revised accordingly. (the study NCT03719690 was added to the table in response to comment originally commenting this study at P10 L350).
The abstract also lists 54 trials (page 1)
4) The authors state “Based on the ongoing studies listed in Table 1 (pages 4-5) and the treatments mentioned in Figure 2 (page 5, line 134), there is no overlap with the new agents that may target the identified genes. This has been emphasized in the discussion (page 11, lines 365-367).”
However, due to the possibly wrong version of the manuscript being uploaded (?) the indicated position P11 L365-367 does not lead to the discussion of novel targets and novel therapies
5) Regarding the critical issue on p53. The authors stated that “Based on our recent publication compiling data from 38 datasets from four databases (Medline, PubMed, Web of Science, and Scopus) involving 8,311 patients from 12 countries, we identified the most prevalent mutated genes in HPV-associated OPC, including TP53 (n = 22), PIK3CA (n = 20), PTEN (n =16), NOTCH1 (n = 14), RB1 (n = 13), FAT1 (n = 13), FBXW7 (n = 12), HRAS (n = 10), KRAS (n = 10), and CDKN2A (n = 10). [15].”
In the new response letter the authors further state “…in https://doi.org/10.3390/biomedicines12010024 ...we showed a supplementary table and a list of all mutated genes with highlighting on the most cited genes…”.
The title of the table is “Supplemental Figure S1: shows list of all genes mutated with highlighting on the most cited genes which are TP53 (n=22), PIK3CA (n=20), PTEN (n=16), NOTCH1 (n=14), RB1 (n=13), FAT1 (n=13), FBXW7 (n=12), HRAS (n=10), KRAS (n=10) and CDKN2A (n=10). Different colour codes representing the 38 articles screened to show which gene was collected from which article.”
Apparently the issue was miscommunication since stating that an analysis of 8,311 patients was performed and finding “TP53 (n = 22), PIK3CA (n=20)…” prevalent mutated genes is not the same as stating we have reviewed 38 studies and 22 of those STUDIES mention or discuss p53 in some context.
P53 is thus not the most prevalent mutated gene (22 patients of 8311 patients) but the gene discussed in most of the reviewed manuscripts (22 of 38 manuscripts). The above must be very explicitly stated as to not leave any room for misinterpretation as there currently is.
Author Response
REVIEWER 2 – second round
In the revised manuscript by de Araújo Ferreira Muniz et al. the authors addressed most of the more technical issues with the original. However, some issues remain or were introduced by the revision.
1) Comment about dose reduced RT (previously at P3L114) remains partially addressed. Indeed, in the revised table 1 now on page 5(?) the authors introduced the RDR and LDR abbreviations as dose reduced and low dose radio therapy. Ie for study NCT01706939. However, for study NCT02254278 the therapy states “Reduced-dose RT“ in full.
More importantly, of 55 currently ongoing studies, only 1 is listed as LDL (NCT04572100) and only 2 are listed as RDR (NCT01706939, NCT02254278). On the other hand, 15 still included only RT (suggestive of not reduced RT).
In light of this, is the following claim still factual: “Most clinical trials involved a reduction in the dose of radiation in combination with chemotherapy, … in an attempt to reduce the side effects (Table 1, Figure 2).“?
Response: The notation “Reduced-dose RT” was replaced by the abbreviation RDR (Table 1, Pages 8-10), to maintain consistency with the rest of the table. As the terms “reduced-dose RT” and “low-dose RT” were used interchangeably in this article, they were standardized to “reduced-dose RT (RDR)” for clarity. Some studies with “RT” interventions did include reduced-dose therapies, but stratified patients for standard or reduced-dose radiotherapy based on their risk level. The interventions for these studies were corrected to reflect the correct dosage (Table 1). After these corrections, 5 studies only used reduced-dose radiotherapy, 8 studies used standard or reduced radiotherapy doses, and 6 studies only included standard radiotherapy doses (Table 1). The claim that most ongoing clinical trials involve reduced-dose radiotherapy was reworded to reflect their use in clinical trials (Page 7, Lines 257-260).
2) The original comment dealing with pages 4 and 5 was partially addressed as well. The authors state “We reformatted the paper to improve the flow of the text and make it clear the paper is exploring existing approved therapies, treatments currently under investigation, and lastly potential treatment avenues that can be further explored in the context of head and neck cancer.” The manuscript indeed has a lot of marked changes, however, the actual order of topics is unchanged from the initial document. At the start of the results (P3L113) the current ongoing studies are presented, followed by lengthy presentation of current treatments and finished by mutations.
Response: The text was reformatted to address this revision. Existing therapies (chemotherapies and immunotherapies) are discussed first, followed by treatments in ongoing clinical trials, then potential treatment avenues for head and neck cancer (Pages 3-7).
Critically, no summary of Table 1 is provided. Figure 2 still does not indicate how many studies actually investigated novel agents (eve if it is one each) or novel combinations (but at least Sintillimab is not erroneously repeated as before). There are 55 studies but only 16 experimental and 8 standard agents listed meaning that a lot of content is not summarized. Please don’t ask readers to browse through Table 1 spanning multiple pages and count how many occurrences of CCDP, TXXT, CBDCA, PTX …etc are there to have a grasp on what is investigated more often and what is not investigated often at this time.
Response: A summary of Table 1 was added to list the number of times each drug was included in current clinical trials, as well as emphasize the most common therapies used (Page 7, Lines 237-260).
Still no textual connection between different aspects of the work is present. No new text was added, only the paragraphs marked as revised.
Response: A sentence that lays out the different aspects that will be covered in the paper was added to the introduction. This should provide clarity to the readers about the rationale and flow of the paper. This addition can be found on Page 2, Lines 68-71.
3) Page 3 L114 “54 of which were related to OPC (Figure 2; Table 1)” is erroneous since Table 1 now contains 55 trials but the sentence was not revised accordingly. (the study NCT03719690 was added to the table in response to comment originally commenting this study at P10 L350).
Response: The number of trials listed was corrected (Page 7, Lines 234).
The abstract also lists 54 trials (page 1, Line 24).
Response: The number of trials listed in the abstract was corrected (Page 1, Line 24).
4) The authors state “Based on the ongoing studies listed in Table 1 (pages 4-5) and the treatments mentioned in Figure 2 (page 5, line 134), there is no overlap with the new agents that may target the identified genes. This has been emphasized in the discussion (page 11, lines 365-367).”
However, due to the possibly wrong version of the manuscript being uploaded (?) the indicated position P11 L365-367 does not lead to the discussion of novel targets and novel therapies
Response: We apologize for using track changes to correct the text, which may have made it difficult to visualize the modifications. To facilitate better visibility, we have highlighted the new reorder in yellow. The discussion of novel targets and novel therapies can be found at Pages 14-15, Lines 429-444. Please note that Figure 2 is now Figure 4 in the revised article.
5) Regarding the critical issue on p53. The authors stated that “Based on our recent publication compiling data from 38 datasets from four databases (Medline, PubMed, Web of Science, and Scopus) involving 8,311 patients from 12 countries, we identified the most prevalent mutated genes in HPV-associated OPC, including TP53 (n = 22), PIK3CA (n = 20), PTEN (n =16), NOTCH1 (n = 14), RB1 (n = 13), FAT1 (n = 13), FBXW7 (n = 12), HRAS (n = 10), KRAS (n = 10), and CDKN2A (n = 10). [15].”
In the new response letter the authors further state “…in https://doi.org/10.3390/biomedicines12010024 ...we showed a supplementary table and a list of all mutated genes with highlighting on the most cited genes…”.
The title of the table is “Supplemental Figure S1: shows list of all genes mutated with highlighting on the most cited genes which are TP53 (n=22), PIK3CA (n=20), PTEN (n=16), NOTCH1 (n=14), RB1 (n=13), FAT1 (n=13), FBXW7 (n=12), HRAS (n=10), KRAS (n=10) and CDKN2A (n=10). Different colour codes representing the 38 articles screened to show which gene was collected from which article.”
Apparently the issue was miscommunication since stating that an analysis of 8,311 patients was performed and finding “TP53 (n = 22), PIK3CA (n=20)…” prevalent mutated genes is not the same as stating we have reviewed 38 studies and 22 of those STUDIES mention or discuss p53 in some context.
P53 is thus not the most prevalent mutated gene (22 patients of 8311 patients) but the gene discussed in most of the reviewed manuscripts (22 of 38 manuscripts). The above must be very explicitly stated as to not leave any room for misinterpretation as there currently is.
Response: The wording of said paragraph was corrected to avoid any confusion and to limit room for misinterpretation. The correction can be found in Section 2.3 Genes involved in HPV-associated OPC (Page 3, Lines 104-112).

Round 3
Reviewer 1 Report
Comments and Suggestions for Authors
All comments were addressed